# Identification of Tumor Antigens and Immune Subtypes of Malignant Mesothelioma for mRNA Vaccine Development

**DOI:** 10.3390/vaccines10081168

**Published:** 2022-07-22

**Authors:** Shuhang Wang, Yuqi Yang, Lu Li, Peiwen Ma, Yale Jiang, Minghui Ge, Yue Yu, Huiyao Huang, Yuan Fang, Ning Jiang, Huilei Miao, Hao Guo, Linlin Yan, Yong Ren, Lichao Sun, Yan Zha, Ning Li

**Affiliations:** 1Clinical Cancer Center, National Cancer Center/National Clinical Research Center for Cancer/Cancer Hospital Chinese Academy of Medical Sciences and Peking Union Medical College, Beijing 100021, China; snowflake201@gmail.com (S.W.); mpw_thumed7@163.com (P.M.); yale_jiang@163.com (Y.J.); yuyueyykk123@163.com (Y.Y.); huanghy314@sina.cn (H.H.); fangchara@163.com (Y.F.); jiangn12@foxmail.com (N.J.); miaohl13@163.com (H.M.); 2NHC Key Laboratory of Pulmonary Immunological Diseases, Guizhou Provincial People’s Hospital, Guiyang 550002, China; 18520224337@163.com; 3State Key Laboratory of Translational Medicine and Innovative Drug Development, Jiangsu Simcere Diagnostics Co., Ltd., Nanjing 210018, China; bioinfo_lilu@163.com (L.L.); minghui.ge@simceredx.com (M.G.); h.guo@foxmail.com (H.G.); linlin.yan@simceredx.com (L.Y.); yong.ren@simceredx.com (Y.R.); 4State Key Laboratory of Molecular Oncology, National Cancer Center/National Clinical Research Center for Cancer/Cancer Hospital Chinese Academy of Medical Sciences and Peking Union Medical College, Beijing 100021, China

**Keywords:** tumor antigens, immune subtype, mRNA vaccine, malignant mesothelioma, tumor immune microenvironment

## Abstract

Background: mRNA-based cancer vaccines have been considered a promising anticancer therapeutic approach against various cancers, yet their efficacy for malignant mesothelioma (MESO) is still not clear. The present study is designed to identify MESO antigens that have the potential for mRNA vaccine development, and to determine the immune subtypes for the selection of suitable patients. Methods: A total of 87 MESO datasets were used for the retrieval of RNA sequencing and clinical data from The Cancer Genome Atlas (TCGA) databases. The possible antigens were identified by a survival and a genome analysis. The samples were divided into two immune subtypes by the application of a consensus clustering algorithm. The functional annotation was also carried out by using the DAVID program. Furthermore, the characterization of each immune subtype related to the immune microenvironment was integrated by an immunogenomic analysis. A protein–protein interaction network was established to categorize the hub genes. Results: The five tumor antigens were identified in MESO. FAM134B, ALDH3A2, SAV1, and RORC were correlated with superior prognoses and the infiltration of antigen-presenting cells (APCs), while FN1 was associated with poor survival and the infiltration of APCs. Two immune subtypes were identified; TM2 exhibited significantly improved survival and was more likely to benefit from vaccination compared with TM1. TM1 was associated with a relatively quiet microenvironment, high tumor mutation burden, and enriched DNA damage repair pathways. The immune checkpoints and immunogenic cell death modulators were also differentially expressed between two subtypes. Finally, FN1 was identified to be the hub gene. Conclusions: FAM134B, ALDH3A2, SAV1, RORC, and FN1 are considered as possible and effective mRNA anti-MESO antigens for the development of an mRNA vaccine, and TM2 patients are the most suitable for vaccination.

## 1. Introduction

MESO is a rare tumor with a highly aggressive biological behavior that primarily originates from the mesothelial surfaces [1], including the pleura (roughly 90% of all MESOs); peritoneum (approximately 10% of all MESOs); or pericardium, tunica vaginalis testis (<1%), after decades of environmental carcinogen exposure, mainly asbestos or asbestos-like fibers [2]. The incidence of MESO continues to increase dramatically worldwide, especially in countries where asbestos is still widely used [2]. Although surgical resection can be offered to patients with early-stage disease, diagnosis is often late for most patients because of its silent clinical course. Systemic chemotherapy with the combination of platin and pemetrexed has been the primary approved frontline therapeutic approach for unresectable MESO in advanced-stage disease for the last two decades [3]. However, the pronounced chemoresistance contributes to limited clinical response [4] and a very poor prognosis, approximately one year with a moderate overall survival (OS), when treated with standard chemotherapy [2]. The worldwide, age-adjusted MESO mortality rate increases by approximately 5.37% annually [5]. Recently, immunotherapy—by combining the CTLA-4 inhibitor and anti-PD-1 antibody—has been approved as a first-line therapy for un-resectable MESO, due to promising clinical results [6,7,8]. However, the limited patient population can benefit from immunotherapy due to the intra-tumoral and inter-tumoral heterogeneity of the MESO tumor microenvironment (TME), including different genomic defects, and different immune landscapes among different patient populations and different areas of the same tumors [9]. Immunotherapy-based strategies still need further attempts to select beneficial MESO patients. Therefore, considering current insufficient therapy options, there remains a crucial unmet necessity for discovering therapeutic targets that are both effective and novel for MESO.

Vaccines against cancer have emerged as another promising immunotherapeutic strategy with the advantages of more specific, persistent immune induction, a broader therapeutic window, minimal toxicity, and producing better therapeutic efficacy, fewer adverse events, non-drug resistance, and lower costs, which are expected but not achieved in standard chemo- and immune- treatments [10,11]. Based on antigen form, the vaccines for cancer are primarily categorized into three broad groups, which include cell (tumor and immune); nucleic acid (RNA, DNA and viral vector); and protein/peptide vaccines [12]. Several vaccines have been studied in preclinical and clinical trials for MESO treatment [13,14,15,16,17]. A listeria monocytogenes vaccine expressing mesothelin (CRS-207) showed a good safety profile, induced positive changes in the tumor microenvironment (TME), and achieved objective tumor responses in 89% of the treated MESO patients with a median 14.7-month OS when combined with cyclophosphamide [13]. TroVax, a viral vaccination containing the 5T4 glycoprotein gene (a highly specific expression in all types of MESO subtypes), combined with first line pemetrexed-cisplatin chemotherapy, has demonstrated robust immune activity, acceptable safety, and tolerability for patients with locally advanced or metastatic MESO in a first-line single-arm phase II trial (SKOPOS) [14]. The median OS was 10.9 months. Even though these vaccines gave a good comparable survival advantage for a few months, the results were quite reassuring for supporting further MESO-related research for the development of vaccines that target a series of self-antigens that are commonly over-expressed in MESO tumors, compared to normal tissues.

Compared to current antigen forms, messenger RNA (mRNA) caused revolutionary vaccine development owing to its potential to induce both humoral and cell-mediated immune responses for high efficacy; an in vivo regulatable and short half-life for a better safety profile; and simplicity and adaptability in antigen design using the same manufacturing platform for a rapid and cost-effective production process [18,19]. Moreover, for mRNA vaccines that are highly suitable for targeting tumor-specific antigens and encoding pathological antigens, sequences of mRNA can be modified and designed [20]. Numerous mRNA vaccines have been investigated for various cancers, showing significant superiority to other types of vaccines. However, neither a MESO antigens mRNA vaccine has been produced, nor sub-population patients been identified that are quite suitable for vaccination.

The objective of the present study was the identification of potential MESO antigens for the vaccine development of mRNA, and to map the MESO immune landscape for the selection of patients that are suitable for the vaccination. Moreover, the tumor antigens that correlated with superior prognoses in MESO were also identified. Similarly, the two immune subtypes were defined by the clustering of the immune-related genes. Furthermore, there was a correlation between the two immune subtypes and differential cellular, clinical, and molecular characteristics, respectively.

In the end, the MESO immunological landscape was described by analyzing the distribution of important gene signatures among the patients. In the present study, a complex TME among different MESO patients was displayed by the analysis. Furthermore, the selection of patients that were suitable for vaccination and the theoretical basis for developing mRNA vaccines were also revealed by the results.

## 2. Materials and Methods

### 2.1. Data Extraction

The RNA-sequencing data and clinical information of 87 MESO tumor samples were collected from The Cancer Genome Atlas (TCGA, https://www.cancer.gov/tcga (accessed on 5 January 2022)). Both microarray and RNA-seq data were standardized using the transformation of log2(x + 1). Furthermore, four immune-related databases, including TISIDB (http://cis.hku.hk/TISIDB/browse.php (accessed on 7 January 2022)); IMMPORT (https://www.immport.org/home (accessed on 7 January 2022)); TIP (http://biocc.hrbmu.edu.cn/TIP/index.jsp (accessed on 7 January 2022)); and Immunome (http://structure.bmc.lu.se/idbase/Immunome/index.php (accessed on 7 January 2022)) were used for the collection of 2526 genes (immune-related) for the study.

### 2.2. Identification of Potential Tumor Antigens

Cancer cells harbor unique mutant genes that theoretically create corresponding unique tumor-specific antigens [21]. Moreover, copy number variation (CNV) burden plays a significant role in tumors’ recurrence and death, indicating that CNV should be considered as an antigen factor [22]. Therefore, we analyzed the mutation frequency and CNV of MESO immune-related genes. Then, we evaluated the associations between these genes and the tumor purity (stromal, immune, and estimate) scores with the “ESTIMATE” R package to attain the tumor purity-related genes [23]. ESTIMATE stands for Estimation of STromal and Immune cells in MAlignant Tumours using Expression data. The ESTIMATE package uses gene expression data to predict the content of stromal and immune cells in malignant tumor tissues. The algorithm is based on performing single-sample gene set-enrichment analysis, producing three scores: stromal score (records the presence of stroma in tumor tissue); immune score (represents the infiltration of immune cells in tumor tissue); estimated score (infers tumor purity). The marker gene sets of the stromal signature include 141 genes; the immune signature also has 141 genes (Appendix A). Finally, we recognized potential genes that were closely related to the MESO progression-free survival (PFS) and OS to obtain candidate genes. The median expression of the candidate genes was used to separate patients into low- and high-expression groups, and the link between the MESO candidate genes and overall survival (OS) was examined in the patients.

The APCs were detected in individual MESO samples using the single sample gene set enrichment analysis (ssGSEA), which examined the relationship between putative tumor antigens and APC infiltration [24]. The R and *p*-values of Pearson correlation were also utilized to determine the association between candidate gene expression and APCs invasion.

Finally, to construct the prognostic index (PI) and identify the most important candidate tumor antigens, multivariate Cox regression analyses were performed.

### 2.3. Immune Subtypes Identification

In the current study, more than two-thirds (1979) of the 2526 TCGA, immune-related genes were mapped to TCGA data. A PAM technique using the “1-Pearson correlation” distance metric was used to partition the immune-related genes after preprocessing. Then, 500 bootstraps were conducted, each including 80 percent of the patients with MESO. Furthermore, a consensus matrix and consensus cumulative distribution function were used to determine the ideal number of clusters, which ranged from 1 to 10.

### 2.4. Cellular and Molecular Characteristics of Immune Subtypes

The tumor immune infiltrating cells (TIICs) and MESO-related genes were discovered to be correlated in the present study analysis. The infiltration score calculations were made using CIBERSORT and ImmuCellAI, a computer program that uses artificial intelligence to determine the number of immune cells in the body [25]. The six immune categories of pan-cancer (C1-C6) and immune-related molecular features were calculated by Thorsson et al. [26]. The whole-exome sequencing (WES) from the TCGA database data was used to describe the mutation spectrum of two clusters to search for specific mutations based on the “maftools” R package.

The tools, i.e., Kyoto Encyclopedia of Genes and Genomes (KEGG) (http://www.genome.ad.jp/kegg/ (accessed on 10 January 2022)) and Gene Ontology (GO) [27,28] are frequently used for the explanation of molecular biology information, including genomic information, protein networks, and biological and gene functions. A functional and pathway enrichment study using the “ClusterProfiler” R package revealed the possible biological importance of DEGs in each of the subtypes with *p* < 0.05 for significance and a cut-off value of |log2 fold-change| ≥ 0.2. The gene set pathway scores for each of the samples were then analyzed using a gene set variation analysis (GSVA). Furthermore, the Broad Institute’s Molecular Signature Database was also used to provide gene sets with c2-curated signatures for the present study. A threshold of |log fold-change| ≥ 0.2 and adjusted *p* < 0.05 (for significance) was used to determine the evident enrichment pathway

### 2.5. Identification of Hub Genes

The STRING database (https://string-db.org/ (accessed on 10 January 2022)) [29] was used to establish a protein–protein interaction (PPI) network. A total of 27 amplified and mutated genes that closely correlated with the OS were included in PPI to identify the hub gene.

### 2.6. Statistical Analyses

All the analyses were performed on R software (version 4.0.5). Differences between the groups were evaluated with categorical variables and continuous data tests by using Fisher and Wilcoxon rank-sum tests, respectively. Immune subtypes were analyzed using multivariate and univariate cox regressions, log-rank tests with covariate stage, and OS outcome to determine their predictive significance. The distinct molecular and cellular features that were associated with different immune subtypes were examined using ANOVA. Furthermore, the chi-square test was used to identify the most commonly altered genes. Similarly, the DAVID tool was used to perform a gene ontology analysis and functionally annotate each gene module. Immune enrichment scores were calculated for each sample using the GSVA package, which is a measure of genes that are up- or down-regulated in tandem in a sample. ssGSEA was employed. K-M plots were developed using the “survival” package. All the analyses were two-sided and *p* < 0.05 was statistically significant.

## 3. Results

### 3.1. MESO Antigens Identification

A total of 24,776 genes that might potentially express tumor-associated antigens were evaluated to discover powerful MESO antigens that could be used in clinical trials. Following that, each patient’s altered genes encoding tumor-specific antigens were screened and 1974 genes were identified. The antigens with immune inhibitory or stimulating properties were identified by screening the genes that had been amplified or mutated. The MESO’s OS was found to be closely linked with 27 genes, whereas PFS was only linked to 12 of the total genes (Figure 1A). The associations between the tumor purity and these 12 immune-survival-related genes were analyzed; the results revealed that four genes, i.e., FAM134B, ALDH3A2, SAV1, and RORC were positively related to tumor purity, and one gene (FN1) was negatively related to tumor purity. These five genes were finally identified as the candidate genes (Figure 2A).

The elevated expressions of FAM134B (Figure 1B); ALDH3A2 (Figure 1C); RORC (Figure 1D); and SAV1 (Figure 1E) were related to superior MESO’s OS, indicating that only four tumor antigens could be assumed to stimulate the immune system. Furthermore, the FAM134B and RORC expression levels were favorably and positively connected with natural killer cells (NK) levels, whereas the AlDH3A2 and SAV1 expression levels were positively correlated with the dendritic cell (DCs) levels, as shown in Figure 2B. In contrast, FN1 genes were associated with a poor OS of MESO (Figure 1F) and negatively associated with the DC and NK cell levels (Figure 2B).

The analysis further revealed a total of tumor antigens including FAM134B, ALDH3A2, SAV1, RORC, and FN1, which were considered as significant candidates for the MESO-mRNA vaccine that could have immune provocative effects and be both presented and processed by APCs for a tumor response induction. Furthermore, using multivariate Cox regression analyses to construct the risk index, we identified two genes among the five, FN1 (HR = 1.22, *p* = 0.014) and RORC (HR = 0.84, *p* = 0.006), that were significantly associated with MESO OS based on the data that were extracted from the TCGA database (Figure 1G).

### 3.2. MESO Immune Subtypes Identification

The tumors and their microenvironment may be mirrored by immune typing, which can assist in the identification of patients who can be good candidates for the vaccination. A total of 1979 immune-related genes were examined in the MESO samples for their expression patterns. Furthermore, eighty-seven tumor samples from four immune-related databases and 282 OS-related genes were chosen to build consensus clustering. The k = 2 was picked where immune-related genes seemed to be stably grouped, based on their cumulative delta and distribution function area (Figure 3A), and generated TM1 and TM2 as two immune subtypes (Figure 3B). TM1 was linked to a worse probability of survival, while TM2 was linked to a better prognosis (Figure 3C). Patients that were identified as differential stages were unevenly grouped based on subtype distribution across distinct tumor stages (Figure 3D).

### 3.3. Immune Subtype Molecular and Cellular Characteristics

We compared significant differences in the tumor purity and immune scores between the two subgroups: patients in TM2 had a higher tumor purity (Figure 4A). Furthermore, the stromal score was significantly lower in TM2; however, the immune score showed no obvious difference (Figure 4B). The same conclusion was proved by the result of ImmuneCellAI (Appendix A).

By using an immunogenomic analysis of more than 1000 tumor samples from 33 cancer types, six immune subtypes were identified, including wound healing (C1); IFN-γ dominant (C2); inflammatory (C3); lymphocyte depleted (C4); immunologically quiet (C5); and TGF-β dominant (C6). In this study, the association between the previously described C1–C6 pan-cancer immune subtypes and two immune subtypes were investigated. The cancers’ immune-modulatory, prognosis, and genetic modifications were strongly linked to these studied groups. Furthermore, the distribution of these six groups was also explored. The analysis further revealed that TM1 and TM2 had a discrete distribution, and the percentage of each immunological category in each of the two immune subtypes was significantly different from one another (Figure 4C,D). For example, wound healing (C1 group) was mainly clustered into TM1, which was reported to be associated with high proliferation, high intra-tumoral heterogeneity, and high adaptive immune infiltration, but less favorable outcomes. On the other hand, IFN-r (C2 group) was mainly clustered into TM2, which had the highest M1/M2 macrophage polarization, a strong CD8 signal, and the greatest diversity of T-cell receptor. In addition, the proportion of C6 (TGF-β dominant) was higher in TM1 than in TM2, which displayed the highest TGF-β signature and a high lymphocytic infiltration. Patients with TM2 tumors, which have a higher level of tumor immunity, were shown to have a longer life expectancy than those with TM1 tumors. In addition to showing the validity of the present study, the immune-typing approach, these data analyses enhance and support earlier categorization.

Furthermore, the immune subtypes and genomic characteristics of the DNA damage repair (DDR) pathways relationship were also observed. A total of five DDR signatures, including intratumor heterogeneity (ITH); copy number variation (CNV); aneuploidy; and homologous recombination deficiency (HRD) were analyzed. It was also found that these pathways of ITH, numbers of segments, and HDR were significantly differently enriched between the two immune subtypes (Figure 4E). Then, the tumor mutation burdens (TMB) of all the samples in TCGA were calculated, and there was no difference between two subtypes (Appendix A).

### 3.4. Immune Modulators and MESO Immune Subtypes Association

The importance of immunogenic cell death (ICD) modulators and immune checkpoints (ICPs) in cancer immunity cannot be ignored. Therefore, in the present study, the expression levels in the different subtypes were analyzed. A total of 25 ICD genes were expressed, of which 6 (24.0%) ICD were differentially expressed among the immune subtypes (Figure 5A). For example, in TM1 tumors, the EIF2AK2, HGF, P2RY2, and PANX1 genes were highly significant and upregulated, while in the TM2 tumors, TLR3 and MET were overexpressed. A total of 45 genes related to ICP (Figure 5B) were discovered and only 13 genes (28.9%) among the immune subtypes were differentially expressed. Furthermore, the ADORA2A, CD44, KIR3DL1, TNFRSF4, TNFRSF9, TNFSF9, TNFSF18, and NRP1 genes were significantly upregulated in TM1 tumors, while CD160, CD200, LAG3, TMIGD2, and TNFSF14 displayed a significantly higher level of expression in tumors (TM2). As a result, it is possible to use immune typing as a biomarker that has the potential use for mRNA vaccines, since it may represent the ICD and ICPs modulators’ expression levels.

### 3.5. Association between Immune Subtypes and Mutational Status

This study found a strong correlation between immune therapy effectiveness, tumor antigen mutational burden (TMB), and mRNA vaccine, which measured the number of tumor antigens in previous research [30]. Since the mutation dataset was mutect2-processed, the TMB and its variants were examined using this dataset/methodology. As shown in Figure 6, the two subtypes showed a significant difference in mutations, including the number and the landscape of mutated genes. The proportion of mutated genes is higher in TM2 tumors (87.5%) than in TM1 tumors (75.86%). Furthermore, the ten immune-related genes landscape with the most mutations that occurred frequently in the genome were significantly different among the two immune subtypes, as shown in Figure 6.

For instance, ALPK3, ARID2, EML5, FAT4, LATS2, TP53, and ZNF469 had specific genomic alteration in TM1 tumors, while ACAP1, CCDC168, DCLRE1B, FANCA, LDB1, and LRP1SETD2 were altered specifically in TM2 tumors. However, TMB among the two immune subtypes showed no significant differences (Appendix A). The analysis in the present study revealed that somatic MESO types and mutation rates in the patients could be easily predicted by the immune subtype.

### 3.6. Function Enrichment Analysis of Immune Subtypes

The GO terms of biological processes (Figure 7A), cellular components (Figure 7B), and molecular functions (Figure 7C) in TM1 tumors were significantly different compared with those in TM2 tumors. Differential expression in various pathways was significantly observed in the two immune subtypes, including antigen processing, angiogenesis, and epithelial mesenchymal transition (EMT).

To further analyze the characteristics of the two immune subtypes, a KEGG functional enrichment analysis was performed with ssGSEA. The TM1 subtypes were highly enriched in various stromal and oncogenic pathways, such as the G2M checkpoint signaling pathway, EMT pathway, hedgehog signaling pathway, hypoxia pathway, antigen processing presentation signal pathway, and VEGF signal pathway (Figure 7D,E and Appendix A).

### 3.7. Identification of Immune Hub Genes

A PPI network analysis was performed to identify the hub regulatory factors. Since FN1 was associated with the most nodes, it was identified as a hub gene (Figure 8). Moreover, FN1 was negatively related with tumor purity (Figure 1G); thus, we finally identified it as a candidate gene, which strongly affects the tumor microenvironment and leads to tumor progression.

## 4. Discussion

Despite the high mortality rate, MESO has just a few treatment options due to its diverse molecular make-up. Combined chemotherapy is the standard treatment for unresectable MESO patients [1], and the immune therapeutic approach is also approved as a first-line therapy [7,8]. With an ideal OS of only 18.1 months, these therapies have dismal therapeutic results. As a result, new, better, and enhanced MESO therapy options need to be explored [6,7]. The mRNA-based vaccines have become a promising immunotherapeutic approach. Nevertheless, complicated tumor heterogeneity hinders the further development of mRNA vaccine-based therapy. Recent studies demonstrated that tumor-associated antigens may be good contenders for a vaccine against cancer through the construction of an aberrantly expressed and mutational landscape for the specific cancer [31,32]. Therefore, the detection and analysis of tumor-associated antigens can identify suitable human mRNA vaccine targets. A vaccine for anti-MESO has yet to be developed based on any detailed investigation of the probable antigens of MESO.

In this study, the abnormal gene expression patterns and mutational landscape of MESO were constructed, and five targeted antigens that had mutations, amplification, and associations with prognosis in MESOs were identified as promising mRNA vaccine candidates, including FAM134B, ALDH3A2, SAV1, RORC, and FN1. The overexpression of the first four antigens was associated with superior OS, PFS, and high APC cell infiltration, while FN1 had a negative correlation. It is worth noting that FN1 was an important oncogene, associated with tumor immune microenvironment. These antigens play an important part in MESO progression and formation, and they may be directly presented and processed to CD8+ T cells to initiate an immunological response. FAM134B, also called JK-1, RETREG1, belongs to a family with sequence similarity 134. Previous studies have indicated that FAM134B may regulate cancer cell death and apoptosis through its role as an endoplasmic reticulum (ER)-phagy receptor, and play specific roles in various malignancies [33]. Interestingly, FAM134B acts as an oncogene in hepatocellular carcinoma (HCC) and esophageal squamous cell carcinoma (ESCC), while it acts as a tumor suppressor in colorectal cancer (CRC) and breast cancer [34]. The mechanism by which it plays different roles in different cancer types has not been thoroughly studied. In CRC, FAM134B can suppress the synthesis phase of the cell cycle to decrease the growth rate and modulate with the Wnt/β-catenin signaling pathway to induce an increase in APC and promote the destabilization of β-catenin, inhibiting the transformation of normal mucosa into adenoma [35,36,37]. In this study, FAM134B may act as the tumor suppressor in MESO, which is similar with CRC. ALDH3A2, a member of the aldehyde dehydrogenase 3 family, is critically important in the detoxification of aldehydes that are generated by alcohol metabolism and lipid peroxidation, and mutations in this gene cause Sjogren-Larsson syndrome [38]. The high expression of ALDH3A2 could improve the prognosis of gastric cancer by remarkably increasing the M1 macrophages to induce an antitumor immune response and kill tumor cells [39]. Therefore, ALDH3A2 was linked to better prognosis. SAV1, expressed in several human cell lines and located in the cell cytoplasm and nucleus, is a core kinase component of the Hippo signaling pathway [40]. It plays an extensive and prominent role in the inhibition of tumorigenesis, mainly through the Hippo pathway in various cancers, including CRC, breast, pancreatic, kidney, lung cancer, and so on [41]. RORC is a member of the nuclear orphan receptor family and performs critical regulatory functions in cell growth, metastasis, chemoresistance, and the negative regulation of PD-L1 expression in various cancers [42,43,44]. Previous studies demonstrated that it could enact these functions through participating in the programmed death ligand-1/integrin β/activator of the transcription 3 (PD-L1/ITGB6/STAT3) signaling pathway in bladder cancer [45]. FN1, a member of the glycoprotein family, is widely expressed by multiple cell types and involved in cellular adhesion and migration processes. FN1 suppressed apoptosis and promoted viability, invasion, and migration in CRC through interacting with integrin α5 (ITGA5) [46]. In vitro study has showed that FN1 upregulation promotes malignant phenotypes in gastric cells by interaction with ITGA5, integrin β1 (ITGB1) and activation of the focal adhesion kinase/steroid receptor coactivator (FAK/Src) axis [47]. To our knowledge, this was the first study that screened the MESO antigens for the development of an mRNA vaccine; however, there is need for further research to determine their mechanisms of action against MESO.

Given that the mRNA vaccine is only beneficial for a fraction of cancer patients due to the tumor heterogeneity, a comprehensive understanding of the immune landscape of MESO is essential for selecting the appropriate patient population. In this study, we classified MESO into two immune subtypes, TM1 and TM2, based on immune gene expression profiles, and they revealed distinct molecular, cellular, and clinical characteristics. The TM2 tumors displayed a prolonged survival compared with the TM1 subtype, which suggested that th immunotype could be used as a prognostic predictor for MESO. Moreover, the TM2 subtypes had higher tumor purity scores and lower stromal scores, which identified a good prognosis of MESO patients in a previous study [48]. Based on previous immunotyping studies, MESO was classified into the C1-C6 subtypes. A different distribution rate of five categories between two subtypes was observed in this study. TM1 mainly overlapped with C1 (wound healing) and C6 (TGF-β dominant) categories, which were associated with an immunologically quiet phenotype and poor prognosis, while TM2 mainly overlapped with C2 (IFN-γ dominant), which was characterized by increased immune-cell infiltration and better prognosis. These six categories represent features of the TME that largely cut across traditional cancer classifications to create groupings and suggest that certain treatment approaches may be independent of histologic type [26], which also demonstrates the superior predictive ability of the immune subtypes in this study over traditional subtypes. The DNA repair processes were frequently found to be altered in MESO, as well as a high degree of intratumor heterogeneity (ITH), copy number of variation (CNV) burden and homologous recombination deficiency (HRD), which demonstrated higher percentages in the TM1 subtype and contributed to poor prognosis and chemoradiotherapy resistance [49]. However, the TMB had no difference, which could be explained by the fact that all of these indications could represent DNA damage, but for different genomic signatures. This result suggested that, for mesothelioma, ITH and HDR may be better indicators of different tumor immune microenvironments. This immune-based classification took not only traditional pathological staging and grading, but also immune-related characteristics into consideration for MESO; in addition, it could distinguish different prognosis well. Therefore, compared to traditional pathological characteristics as a single-level consideration, immunotyping is a comprehensive prognostic predictor combing conventional clinical-related and novel immune-related characteristics for MESO.

In addition to prognosis, immunotype can also indicate the therapeutic response to an mRNA vaccine. First, an mRNA vaccine could be more effective for patients with highly expressing immunogenic cell death modulators, while it may not suitable for those with an upregulation of immune checkpoints, which may inhibit the vaccine from eliciting an effect immune response. Second, TM2 tumors with higher somatic mutation rates may have greater responsiveness to an mRNA vaccine. Third, the TM1 subtype was highly enriched in various stromal and oncogenic pathways, which indicated that patients in the TM1 subtype might not be suitable for the mRNA vaccine. Finally, FN1 was identified as the hub gene, and the upregulation was negative with tumor purity, suggesting that patients expressing high levels of FN1 may not respond to the mRNA vaccine. FN1 could be a potential biomarker for an mRNA vaccine. However, our study was limited because it was retrospective, and the vaccine antigens and other prognostic markers that were identified need to be validated. Therefore, further in vitro and in vivo studies should be explored to evidence the positive impact of reinforcing the selections of these identified tumor antigens for an mRNA vaccine in the future.

## 5. Conclusions

In conclusion, for vaccine development, the FAM134B, ALDH3A2, SAV1, RORC, and FN1 genes are potential MESO antigens for mRNA and could be beneficial for TM2 patients. Most importantly, FN1 is a potential biomarker for the mRNA vaccine as it is a hub gene. Targeting FN1 may prove the efficacy of the mRNA vaccine. Furthermore, in the present study, an mRNA vaccine for MESO is theoretically possible and identifies which patients are most suited for vaccinations.

## Figures and Tables

**Figure 1 vaccines-10-01168-f001:**
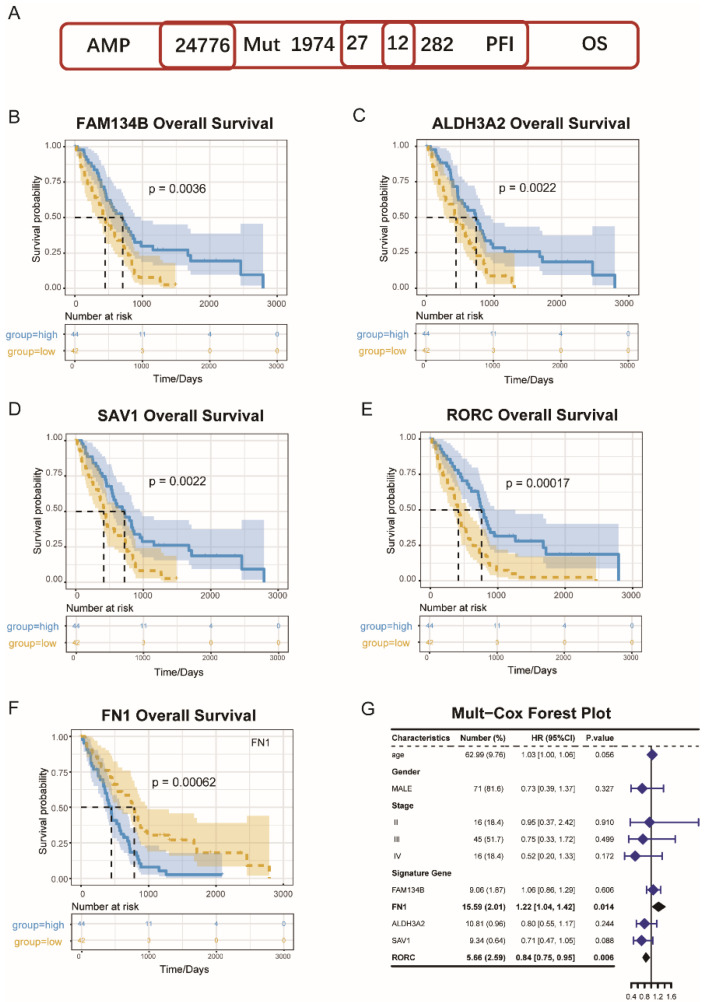
Identification of potential tumor antigens of MESO. (**A**) Potential tumor antigens with high expression and mutation in MESO, and significant association with OS and PFI. B-F. Kaplan-Meier curves showing OS of MESO patients stratified of the basis of (**B**) FAM134B, (**C**) ALDH3A2, (**D**) SAV1, (**E**) RORC, (**F**) FN1 expression levels. (**G**) Mul-Cox Forest Plot for OS of MESO.

**Figure 2 vaccines-10-01168-f002:**
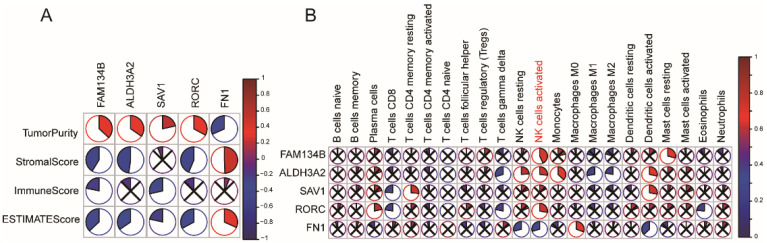
Identification of tumor antigens associated with tumor purity, immune scores (**A**) and APCs (**B**).

**Figure 3 vaccines-10-01168-f003:**
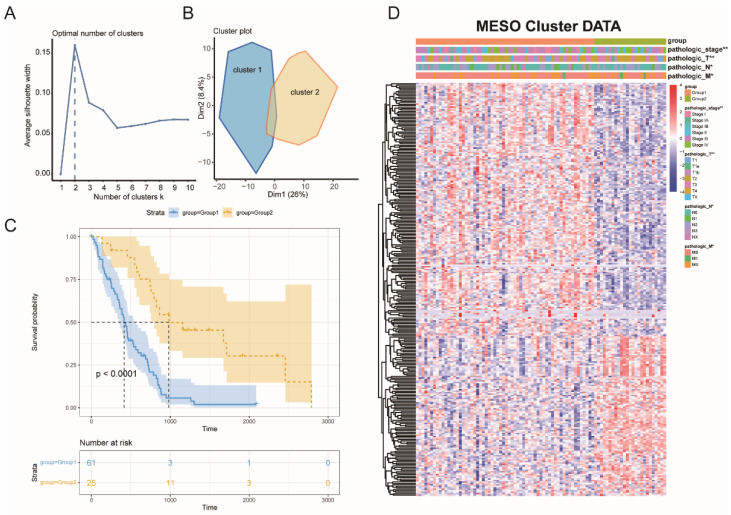
Identification of potential immune subtypes of MESO. (**A**) delta area of immune-related genes. (**B**) Sample clustering heat map. (**C**) Kaplan-Meier curves showing OS of MESO immune subtypes. (**D**) Distribution of TM1 and TM2 across stages. * *p* < 0.01, ** *p* < 0.001.

**Figure 4 vaccines-10-01168-f004:**
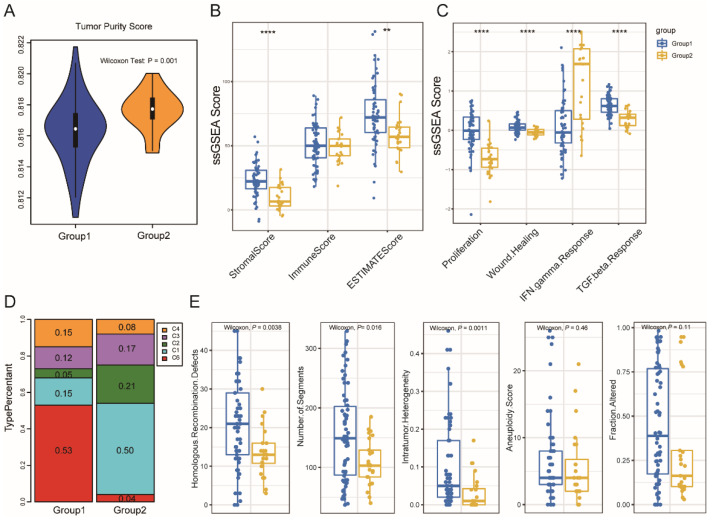
Cellular and molecular characteristics of immune subtypes. Tumor purity score (**A**) and Immune score (**B**) of the two immune subtypes. of the two immune subtypes. (**C**,**D**) Distribution of individual immune categories in the two immune subtypes. (**E**) Genomic characteristics of DNA damage repair pathways in the two immune subtypes. ** *p* < 0.001 and **** *p* < 0.00001.

**Figure 5 vaccines-10-01168-f005:**
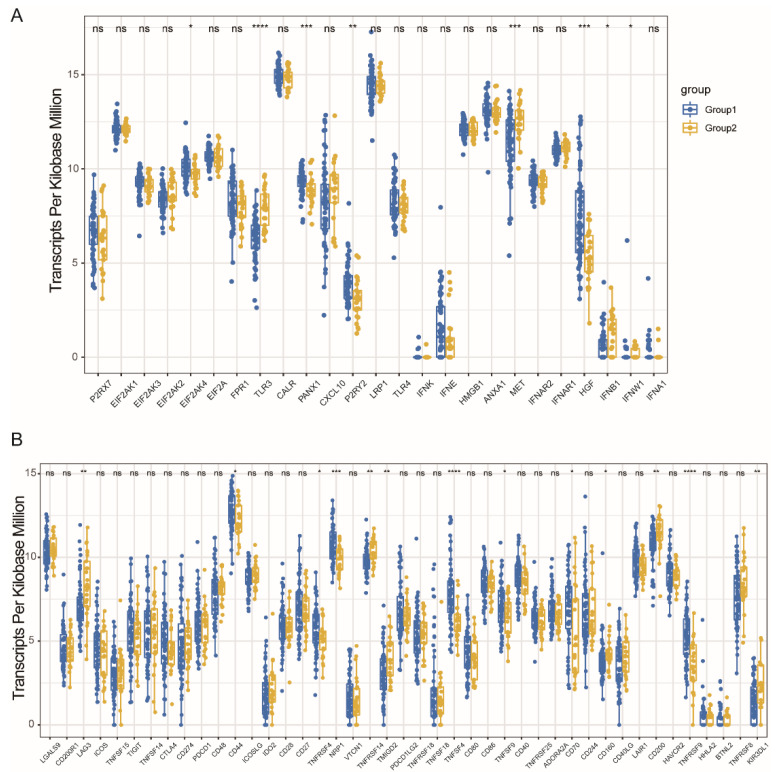
Association between immune subtypes and ICD modulators and ICPs. Differential expression of ICD modulator genes (**A**) and ICP genes (**B**) between MESO immune subtypes. * *p* < 0.01, ** *p* < 0.001, *** *p* < 0.0001, and **** *p* < 0.00001. ns: no significance.

**Figure 6 vaccines-10-01168-f006:**
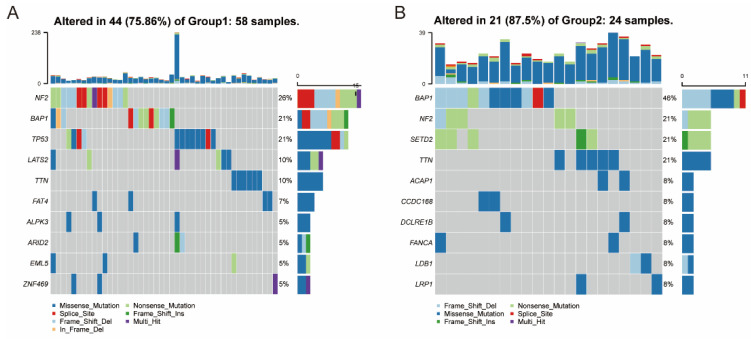
Association between immune subtypes and mutation. The landscape of the genomic alteration of 10 representative immune-related genes in TM1 subtype (**A**) and TM2 subtype (**B**).

**Figure 7 vaccines-10-01168-f007:**
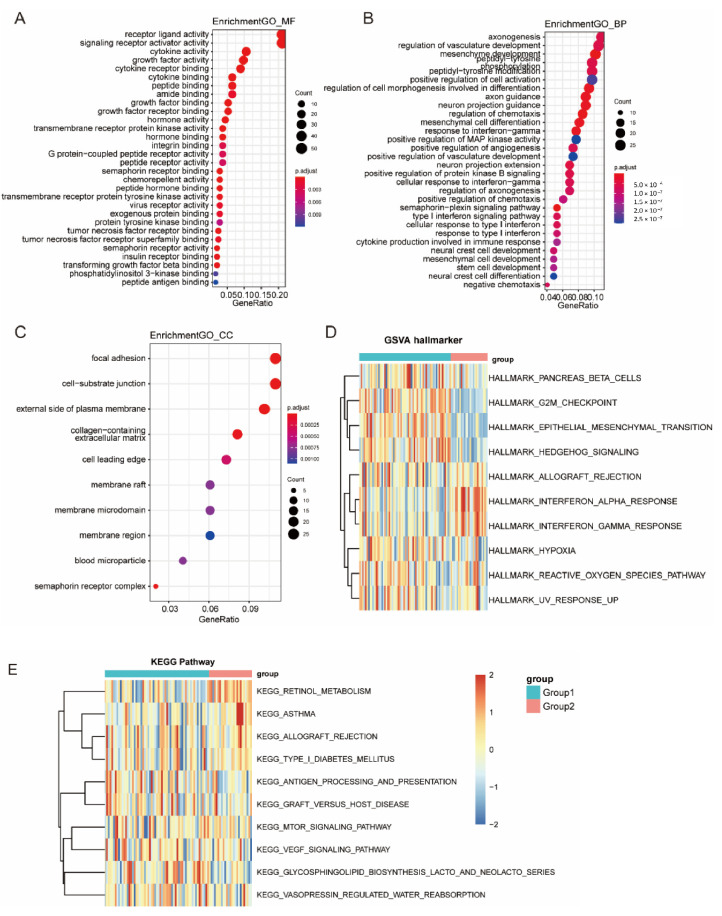
Function enrichment analysis of immune subtypes. (**A**–**C**) GO enrichment analysis in the two immune subtypes, including MF: molecular function (**A**), BP: biology process (**B**), CC: cellular component (**C**). (**D**,**E**) KEGG enrichment analysis in the two immune subtypes, including enriched GSVA hallmarkers (**D**) and KEGG pathways (**E**).

**Figure 8 vaccines-10-01168-f008:**
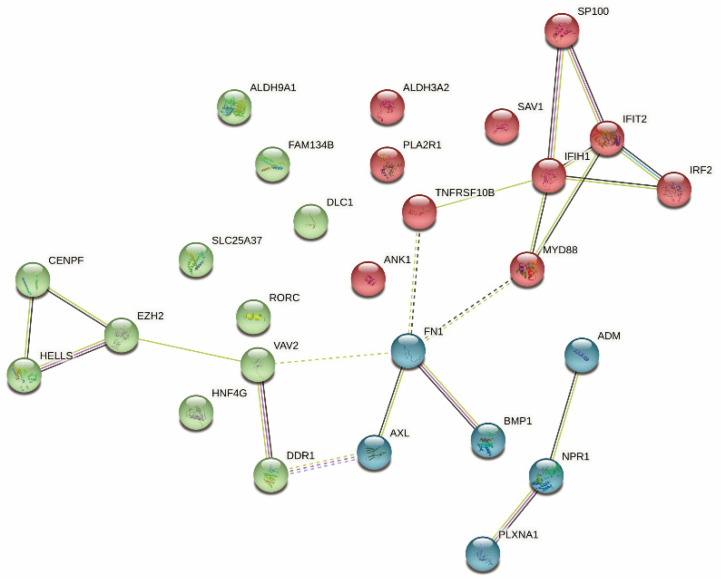
Identification of hub genes of MESO.

## Data Availability

All data generated and/or analyzed during the current study are available from the TCGA datasets.

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
