# Peer review of "Identification of Tumor Antigens and Immune Subtypes of Malignant Mesothelioma for mRNA Vaccine Development"

_vaccines, 2022, doi:10.3390/vaccines10081168_

Round 1
Reviewer 1 Report
Dear authors,
Thanks for your contribution on this field. This is an interesting article on identifying tumor antigens for mRNA vaccine development in the case of mesothelium cancer. The manuscript is well organized and written. Research is well conducted.
I can raise some concerns about figure quality which render them in some case unreadable.
Even if some robust data are provided on the identity gene are evidenced, all result are coming from data mining. It will be more convincing if some in vitro data from cell culture-based experiments are provided such as identification by qPCR or Western blot of identified target and such as silencing experiments of their expression aiming at evidencing the positive impact to reinforce their selections as tumor antigens for mRNA vaccine.
Regards
Reviewer 2 Report
This paper is an in-silico search for antigens that could be used to improve responses to treatment of mesothelioma. Part of the analysis involves breakdown of MESO into 2 immune subtypes, TM1 and TM2 with a corresponding prognosis and for TM2a list of antigens that could be considered for immune antibody therapy.
There are several issues, both minor and major that should be addressed before publication.
There seems to be a discrepancy over mutation burden in TM1 versus TM2, that arises because different data sets are used for analysis. In Figure 4, TM1 shows greater DNA alterations in the first 3 panels. However, in sup Fig B, the mutational burden is not different. Also, put the P value in the panel. Please, point out in the Results section and discuss after. Also, state clearly how the Tumor purity scores are calculated so that we can see how the StomaSscore seem to be strong enough the give a significant difference in the EstimateScore in Fig 4 B even though the ImmuneScore is not different. Also, provide the markers for StromaSscore.
Line 83-MPM; please spell out rather than using an abbreviation.
Line 113-say “87 MESO tumor samples”
Lime 123- reverse, copy number variation (CNV)
Line 189-genes identified
Line 196-, which was
Line 205- natural killer cells
Line 217-218. The entire sentence should go to the Discussion section, line 351 or near there.
Line 226- eighty-seven tumor samples
Line 233-234- Provide a comparison to support claim. State here that this will be included in the Discussion section. There, provide the comparable information from the literature with references.
Figure 3B. Change colors of clusters to agree with 3C and 4 and 5. Cluster 2 should be orange and cluster 1 should be blue.
Lines 252-257. List out all 6 groups. Also, tells us what “wound healing” means for cancer. Could this be rephrased as metastatic potential?
Line 302. “ had” instead of “detected”
Line 034. Need a verb, LRP1SETD2 “were”
Line 324. (hub genes)
Line 328. Affects
Line 368. ALDH3A2 was linked to better prognosis
Line 382. In vitro
Line 409. to prognosis,
Line 411-412. Spell out ICD and ICP
Reviewer 3 Report
This article investigated that the possible candidates of antigen for mRNA vaccine against malignant mesothelioma by in silico analyses. Authors drew a reasonable conclusion from the obtained results.
The contents are well organized, however, the following point should be considered to improve this article.
Minor point
1) There are some parts that require minor corrections.
a) Figure4 A : Wilcox Test → Wilcoxon Test
b) Line 311 : patways → pathways
c) Line 499 : Elife → eLife
d) Line 539 : EBioMedicine → eBioMedicine
2) The expression style should be unified.
In “References” section
a) for example [1] vs [3] : Wang S vs Kindler H.L.
b) CA : Cancer Journal for Clinicians → CA Cancer J Clin
Author Response
Response to Reviewer 3 Comments
Point 1: There are some parts that require minor corrections.
- a) Figure4 A : Wilcox Test → Wilcoxon Test
- b) Line 311 : patways → pathways
- c) Line 499 : Elife → eLife
- d) Line 539 : EBioMedicine → eBioMedicine
Response 1: Thanks very much for your corrections. I’m very sorry that we had some spelling mistakes and we have corrected them in the reversed manuscript.
2) The expression style should be unified.
In “References” section
- a) for example [1] vs [3] : Wang S vs Kindler H.L.
- b) CA : Cancer Journal for Clinicians → CA Cancer J Clin
Response 2: Thanks very much for your corrections. We have checked all the references and unified the expression style in the reversed manuscript.
Round 2
Reviewer 1 Report
Thanks for considering the made comments.
I can understand the difficulty of getting patient samples for such rare cancer.
Hope you will be able to provide robust data on next papers
All the best
Reviewer 2 Report
Do one more check for English/grammar.